# The Effect of a Brief, Web-Based Animated Video for Improving Comprehension and Implementation Feasibility for Reducing Anterior Cruciate Ligament Injury: A Three-Arm Randomized Controlled Trial

**DOI:** 10.3390/ijerph18179092

**Published:** 2021-08-28

**Authors:** Erich J. Petushek, Anne Inger Mørtvedt, Brittany L. Nelson, Mary C. Hamati

**Affiliations:** 1Department of Cognitive and Learning Sciences, Michigan Technological University, Houghton, MI 49931, USA; amrtvedt@mtu.edu (A.I.M.); bnelson1@mtu.edu (B.L.N.); 2Health Research Institute, Michigan Technological University, Houghton, MI 49931, USA; 3Department of Orthopedics, School of Medicine, University of Colorado, Aurora, CO 80045, USA; mchamati@outlook.com

**Keywords:** multimedia, learning, motivation, uptake, education, information, knee, coach, sport, behavior change

## Abstract

Neuromuscular injury prevention training (IPT) has been shown to reduce anterior cruciate ligament (ACL) injury risk by approximately 50%, but the implementation rate is low. One of the most important modifiable barriers for implementation is coaches’ comprehension of risk and intervention strategies. This study aimed to evaluate the effect of a brief, web-based, animated video on ACL injury prevention comprehension and IPT implementation feasibility. Coaches in landing and cutting sports were recruited and randomized into three groups. (1) Intervention: brief multimedia animated video about ACL injury and prevention. (2) Active control: commonly accessed, text-based web resource about ACL injury and prevention. (3) Placebo control: brief multimedia video about concussions. Overall ACL comprehension—composed of basic ACL knowledge, risk knowledge, prevention knowledge, and severity knowledge—as well as implementation feasibility were all measured prior to and immediately following the interventions. Overall ACL comprehension improved the most in the animated video group (Cohen’s *d* = 0.86) and, to a lesser degree, in the active control web-based article group (Cohen’s *d* = 0.39). Both video and web-based article groups had greater implementation feasibility compared to the control group (*p* = 0.01). Overall, these initial results suggest that a brief, web-based, animated video has the potential to be a superior method for informing stakeholders in order to reduce traumatic injuries in sport.

## 1. Introduction

Anterior cruciate ligament (ACL) injuries remain a severe concern among female athletes participating in landing and cutting sports such as basketball, soccer, and volleyball [1]. Even though implementation of neuromuscular injury prevention training (IPT) has been shown to reduce the injury risk by approximately 50% [2,3,4,5], the implementation rate is low (~4%–20%) [6,7,8]. Previous studies indicate that the key to the implementation of IPT is through increasing knowledge and comprehension of the injury and evidence-based prevention strategies among coaches [9,10,11].

Previous studies have reported different barriers for implementation of IPT programs, with coaches’ comprehension and knowledge of intervention strategies as the most important modifiable barrier [7,10,12,13,14,15]. Coaches represent the primary and most efficient stakeholder to implement IPT [16], but coaches’ knowledge of IPT lags behind the knowledge of sports medicine professionals [17]. Hence, scientists and practitioners have advocated education/training for coaches to increase IPT usage [9,13,18,19]. Large-scale implementation efforts are needed, especially to target the population of adolescent athletes, which often do not have sports medicine professionals affiliated with their team. To be able to reach a larger number of coaches, athletes and other stakeholders, more efficient, accessible, and enjoyable methods for improving comprehension and motivation to implement ACL IPT are needed.

Current research on ACL IPT education has relied on relatively long, resource-intensive in-person sessions, and no study has utilized a brief web-based delivery to improve ACL injury risk and mitigation comprehension. Web-based methods have the potential to reach audiences all over the world, are cost-effective, and are less time-consuming. Video interventions provide the ability for multisensory learning [20], reduce workload [21], and increase motivation to view and/or learn [22]. Video-based interventions are also valuable for educating groups with limited health literacy, and/or low numeracy [23,24]. Strong evidence suggests that videos increase informed decision-making [25], are effective for learning (e.g., [26,27,28,29,30]), and increase both short- and long-term retention [31].

Video-based training and education have been shown to be effective for athletes [32,33] and referees [34,35]. However, few studies have applied this approach to coaches, and when carried out, are often limited to the concussion domain where education is sometimes legislatively mandated [36,37]. Even in the concussion domain, video and internet-based training methods have received little attention and long-term behavior change has yet to be rigorously tested [38,39]. Some key findings from the limited evidence on video-based methods point to the benefits of presenting factual information clearly, focusing on the personal and contextual factors as well as optimizing enjoyability.

The aim of this study is to evaluate the effects of a brief web-based educational intervention on improving ACL injury prevention comprehension and implementation feasibility among coaches of athletes participating in landing and cutting sports (e.g., soccer, basketball, football, etc.). Our two main hypotheses are that, compared to both an active and placebo control group, the brief animated video will produce greater improvements in:Overall comprehension of ACL injury risk and mitigation: placebo control group < active control group < intervention group.Feasibility of utilizing ACL injury prevention strategies: placebo control group < active control group < intervention group.

Exploratory hypothesis: The brief animated video will produce greater improvements in various subcomponents of comprehension—specifically: basic ACL knowledge, risk knowledge, prevention knowledge and severity knowledge compared to both active and placebo control group conditions.

## 2. Materials and Methods

### 2.1. Trial Design

This study was a computer-based, three-arm pre–post randomized control design trial. Participants were randomized into an intervention group, active control group or passive control group on a 1:1:1 ratio by a computer-generated algorithm embedded within the Qualtrics software (Qualtrics, Provo, UT, USA). The study was approved by the Michigan Technological University institutional review board (project number 1521444-1).

### 2.2. Procedure

All participants conducted an online survey using the Qualtrics platform, which consisted of various questions to assess comprehension of ACL injury risk and mitigation, the feasibility of implementing prevention strategies, and demographic questions. Participants were initially prompted with a consent page where they were provided with brief information about the purpose of the research project (i.e., to learn about sports injuries, that their data would be anonymous, time requirements) and were asked to consent to participate in the project. Participants who consented then proceeded to the survey for eligibility screening and concealed allocation, central randomization by Qualtrics software. The Randomizer function in Qualtrics (simple randomization) was used to block randomize participants into three even groups: intervention (animated video), active control (text-based web article) and placebo control. The comprehension questions were repeated to all groups immediately following the interventions.

### 2.3. Participants

Sample size (75 in each group) was determined based on an estimated moderate effect size (*d* = 0.50), alpha of 0.01 and power of 85%. The effect size estimate is based on our previous cross-sectional studies of differences in ACL risk estimation skill and knowledge, as well as previous education interventions [17,18]. The alpha level was adjusted based on the five outcome measures. The power was chosen based on recent concerns about replication and inadequate power [40,41,42,43]. Sports coaches of youth and adolescent athletes in various landing and cutting sports (e.g., soccer, basketball, volleyball, etc.) in the United States were recruited and invited through Qualtrics’ panelists in the period between November 2019 and June 2020. Qualtrics panels have been shown to be among the best tools for gaining access to representative and non-probability samples [44]. Participants were invited, through email, based on their potential demographics (e.g., sport coaches) and screened based on the sports they coached. Participants needed to be 18 years of age or older. Participants received a small monetary compensation for completing the survey.

### 2.4. Intervention

The intervention group was shown a three-minute animated video consisting of various information components aimed at improving capability, motivation and opportunity to implement ACL injury prevention strategies. The information in the ACL animated video (https://vimeo.com/281721823; accessed on 1 October 2019) was displayed as a story of a typical athlete who sustained an injury and how this could be prevented through evidence-based prevention strategies. The video content/script was developed in conjunction with sports medicine professionals, psychology researchers, athletes, coaches and Kindea Labs (a film/video production company). The information presented about the prevention strategies was based on up-to-date meta-analytic evidence of the essential components for reducing ACL injuries [3]. The organization followed similar strategies to that of in-person workshops (e.g., the problem/epidemiology and solution/prevention strategies). The video was pilot/usability tested with five coaches and athletes to remediate any issues of misunderstanding and confusion. The active control group received commonly accessed information from a WebMD web-based article on ACL injury prevention (see Appendix A [45]). The information presented in the web-based article aligns with evidence-based practice recommendations [3]. No illustrations or figures were included in the active control group. Both the intervention group and the active control group received ACL-related information including the purpose of the ligament, injury mechanisms and severity, risk factors and proposed exercises to reduce injury risk. The placebo control group intervention received an educational video from the CDC about concussions that is comparable in duration to that of the ACL video (https://youtu.be/fSRWF44wgn8; accessed on 1 October 2019).

### 2.5. Outcome Measures

Items that measure ACL knowledge were gathered from peer-reviewed, published articles [10,46,47]. Together, these items measured overall ACL injury comprehension. Previously used ACL knowledge assessments have documented convergent validity evidence (e.g., see [46]). According to the items’ content, the items were also grouped together to measure four subcomponents of ACL injury comprehension: basic ACL knowledge, risk knowledge, prevention knowledge, and severity knowledge (see Appendix B for full item listings). Basic ACL knowledge items were designed to measure what ACL stands for, as well as the location of the ACL. Risk knowledge items were designed to measure the perceived risk of ACL injuries for age and gender demographic groups. Prevention knowledge measured understanding of how to reduce the risk of ACL injuries. Finally, severity knowledge measured understanding of the long-term repercussions of ACL injuries. Pearson product-moment correlation coefficients were calculated on participants in the control group (N = 86) to determine test–retest reliability across the four knowledge factors. Results showed good to excellent correlations (r > 0.67, *p* < 0.01), suggesting these measures have strong test–retest reliability and that the participants were taking these questions seriously [48]. The feasibility of the intervention measure was used to assess the likelihood that the coach would successfully implement evidence-based prevention strategies. This 4-item measure has been shown to have sufficient test–retest reliability (r = 0.88) and overall structural validity [49]. The 4 items consisted of the root phrase: “An ACL injury prevention program seems” followed by (1) implementable, (2) possible, (3) doable and (4) easy to use. The items were rated on a 5-point Likert scale ranging from completely disagree to completely agree. The average of the 4 items was used for the analysis. Participants responded to the implementation feasibility only following the interventions, as it was unlikely that participants would be aware of such prevention programs prior to the information presented.

### 2.6. Statistical Analyses

Participants who responded with incoherent or implausible answers to the demographic questions or attention checks were screened by a trained rater prior to randomization. Initial one-way ANOVAs were used to determine any baseline differences in comprehension outcome variables. To test whether there were significant differences between conditions on overall ACL injury comprehension, a mixed ANOVA was used to analyze the effect of two levels of the repeated measure variable of time (pre-test versus post-test) and three levels of the between-group condition (animated video, control, and web-based article). To identify if differences existed on the four subcomponents—basic ACL knowledge, risk knowledge, prevention knowledge, and severity knowledge—four separate additional mixed ANOVAs were performed. The magnitude of differences between conditions pre-test versus post-test and the between-group condition (animated video, control, and web-based article) were measured using *t*-tests and Cohen’s d. Differences in ACL implementation feasibility between the animated video intervention compared to the active and placebo control conditions were analyzed using a one-way ANOVA, as this variable was only measured post-intervention. Cohen’s d was also calculated to measure the magnitude of differences between groups for the implementation feasibility variable. Multiple comparison *p* values were corrected using the Benjamini–Yekutieli adjustment [50]. All analyses were conducted using RStudio [51].

## 3. Results

Four-hundred and seventy-nine participants (animated video N = 169, control N = 163, web-based article = 147) started the survey. Two-hundred-twenty-three were excluded due to incoherence, implausibility, or not meeting the inclusion criteria based on sport, resulting in 256 participants eligible for randomization and analyses (see the flow chart in Figure 1). Table 1 reports the demographic characteristic of the analyzed sample. No differences were found between groups for any of the demographic variables (all *p* > 0.13). The majority of the coaches in the sample were male, 25–44-years-old, had roughly 7 years of experience, and coached soccer, basketball, and football. Only roughly 15% of the sample were aware of ACL injury prevention programs prior to the interventions.

**Hypothesis** **1.**
*The ACL video educational intervention will produce greater improvements in overall ACL comprehension (i.e., basic ACL knowledge, risk knowledge, prevention knowledge, and severity knowledge) compared to both the active and placebo control group conditions.*


Results from the one-way ANOVA suggested there were no significant differences between groups at time 1, on overall ACL comprehension or the four knowledge components (*p* > 0.56). Table 2 shows the results of the mixed ANOVA comparisons. The results show a significant interaction and a significant main effect on overall comprehension. Further analyses were conducted to identify which knowledge components changed the most. Results show significant interactions for two of the knowledge components (risk knowledge and severity knowledge) between pre and post tests and experimental conditions (*p* < 0.001). Results also show a significant main effect across two of the knowledge components (risk knowledge and severity knowledge) (*p* < 0.001).

Cohen’s d was calculated to compare differences between conditions and time (see Table 3 and Table 4). Table 3 shows the means and standard deviations of the pre and post test results for the animated video, web-based article, and control-group conditions. Negligible effect sizes were found in the control group condition across the four knowledge factors. The web-based article group had negligible to moderate effect sizes across the conditions. With the exception of basic knowledge (where the difference was 0.01), the results show the largest effect sizes for the animated video condition. Small to moderate effect sizes were also found between time 1 and time 2 on the animated video condition; the effect sizes were larger compared to the active and placebo control groups, with the exception of basic ACL knowledge (see Table 4). Figure 2 shows the difference in percent correct between time 1 and time 2 on overall comprehension by condition.

**Hypothesis** **2.**
*The animated video will produce greater improvements in feasibility to utilize ACL injury prevention strategies compared to both active and placebo control group conditions.*


Table 5 shows the means and standard deviations of the animated video, active and placebo control group conditions for the feasibility of implementation. Results show a statistically significant difference between animated video, active and placebo control group conditions on the feasibility to utilize injury prevention strategies, with the animated video having the greatest feasibility for implementation (F(2, 253) = 4.45, *p* = 0.01). A follow-up independent t-test revealed a statistically significant difference between the web-based and placebo control group conditions (Table 5; *p* = 0.03). There was also a statistically significant difference between the control and ACL video conditions (Table 5; *p* = 0.03). Corresponding *p* and Cohen’s d values between the intervention groups are in Table 5.

## 4. Discussion

The aim of the current study was to evaluate the acute effects of a brief, newly developed animated video on ACL injury prevention comprehension and implementation feasibility. The results indicate that the use of a brief animated video significantly improves overall ACL injury comprehension compared to both the active control group (web-based article) (*d* = 0.49) and the placebo control group (*d* = 0.56). Analysis of the four separate knowledge areas of ACL injury prevention comprehension revealed the greatest improvement within risk knowledge and severity knowledge. The improvement in prevention knowledge was also greater for the animated video group compared to both the control and the web-based article group, but the effect size was smaller than for risk and severity knowledge. No differences were found between the groups for basic ACL knowledge. This finding may be due to a ceiling effect since this component consisted of two questions that the majority of participants answered correctly. These results suggest that the animated video has an immediate impact on increasing overall ACL injury comprehension with the greatest difference found on the risk and severity knowledge subcomponents.

The increased effectiveness in improving comprehension through visual aids and multimedia approaches reported in the current study is supported by skilled decision theory [52] and the cognitive theory of multimedia learning [53]. The animated video was also designed as a narrative, which can have a positive effect on knowledge, attitudes and health behaviors [54]. The theoretical foundation behind the current study is that representative comprehension is fundamental for making skilled decisions [52]. Therefore, increasing comprehension of ACL injury prevention among important stakeholders for implementation of IPT in high-risk sports will potentially increase the likelihood of IPT adoption and skilled implementation. Using an animated video to increase comprehension of ACL IPT proved to be the most effective information format in the current study. However, even though both the animated video group and the active control group showed greater implementation feasibility compared to the placebo control group, the difference between the animated video and active control group was smaller than hypothesized. The questions addressing implementation feasibility in the current study may not have been able to detect differences between groups due to a high number of maximum scores, causing a ceiling effect [55]. Future studies would benefit from recording actual uptake of prevention behaviors in addition to acute intentions or feasibility.

There is convincing evidence that IPT is effective for reducing risk for several sports-related injuries [2,3,4,5,56], but the translation from research to practice, including risk communication between researchers and different stakeholders for adoption of IPT, is insufficient. Low sensitivity to contextual factors and the actual level of comprehension required have been proposed as reasons for the lack of uptake [57,58]. Videos have been shown to be more effective for both learning, motivation and short- and long-term retention, which may reflect some of the differences seen between the animated video and the web-based article group in pre- and post-analyses on comprehension. For any behavior to occur, people need to have the capability, opportunity and motivation to engage in it, and all three conditions influence each other [59]. The current study showed that interventions had an effect on comprehension as well as the feasibility for ACL IPT implementation, which could be a surrogate for motivation to implement IPT. Engaging in a behavior demands that the person concerned is more motivated to enact this behavior than enacting anything else in the moment where the behavior occurs. This means that performing IPT would have to overcome every other behavior coaches might want their athletes to engage in and highlights the importance of improved risk communication for improving implementation feasibility and compliance of ACL IPT [59].

Many studies have been conducted to determine specific exercises and effective components of injury prevention programs [3,5,8,18]; however, there has been less focus on the role of ACL IPT education as an important part of IPT adoption at the team level. A previous study, which examined the effects of a workshop for coaches, reported that the participants significantly increased their attitudes towards, confidence in and intention to implement IPT the next season, and more than half of the teams actually did implement IPT the following season [18]. Among these teams, high variability in fidelity was reported. Their results indicate that in-person workshops may be an effective educational approach for increasing the adoption of ACL IPT, but the need for an intervention that reaches a larger number of stakeholders is crucial. The development of a web-based educational platform may be an effective method to reach a larger scale of important stakeholders through higher accessibility and intervention efficiency with less variability in the content delivered. Providing stakeholders with easy access to a brief online video will also make them able to repeat and explore all or parts of the content as needed or preferred. The results of the current study indicate that the brief animated video is effective in improving comprehension, is cost-effective and has an effect on implementation feasibility, but the effects on actual implementation rates and fidelity are yet to be tested. Despite the improvements in overall comprehension, there is still room for further improvement as the average percent correct following the animated video was still 44%. Thus, future research is needed to improve the animated video content to address these shortcomings, and particularly in prevention and severity knowledge.

Regarding limitations, as discussed above, the implementation feasibility variable may have been victim of a ceiling effect due to the high number of maximum scores, making it likely that analyses were unable to detect differences between groups. Longer follow-up or delay periods may also improve the design and results to mitigate reduce recall bias. Another potential limitation in this study is that the quality of the content delivered through the animated video and the web-based article was different. Even though both information formats included information about injury mechanisms and severity, risk factors and proposed exercises and prevention strategies for ACL injuries, the content may have been different in more ways than a web-based article versus video animations (using different terms, explanations, etc.). The results may have been different if another text-based intervention were chosen. The rationale for selecting the WebMD web-based article was to resemble typical health information found on the internet. In the US home market, WebMD and NIH.gov have the highest number of monthly visitors of all general-purpose health information sites, and thus was representative of an “existing practice or active control” group [60]. Unfortunately, many of the web-based resources related to ACL injury prevention have low usability, readability, and vary in content [61,62]. A final limitation was the short-term nature of the follow-up/post-test. Future studies would benefit from assessing retention or longer-term follow-up to see if knowledge was retained.

## 5. Conclusions

The results of the current study indicate that the use of an animated video is effective for acutely increasing comprehension of ACL IPT and that this education method is superior to information presented on routinely visited health information websites (e.g., WebMD). Increased comprehension is an important part of behavior change, but further studies are needed to examine the effects of increased comprehension on short-term and long-term IPT compliance and exercise fidelity.

## Figures and Tables

**Figure 1 ijerph-18-09092-f001:**
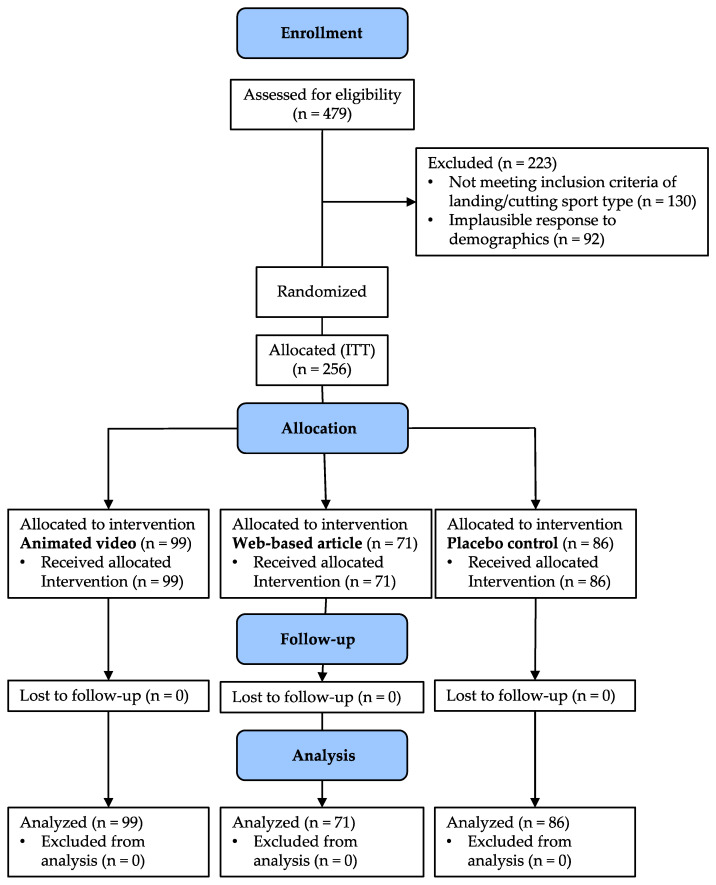
Participant flow chart.

**Figure 2 ijerph-18-09092-f002:**
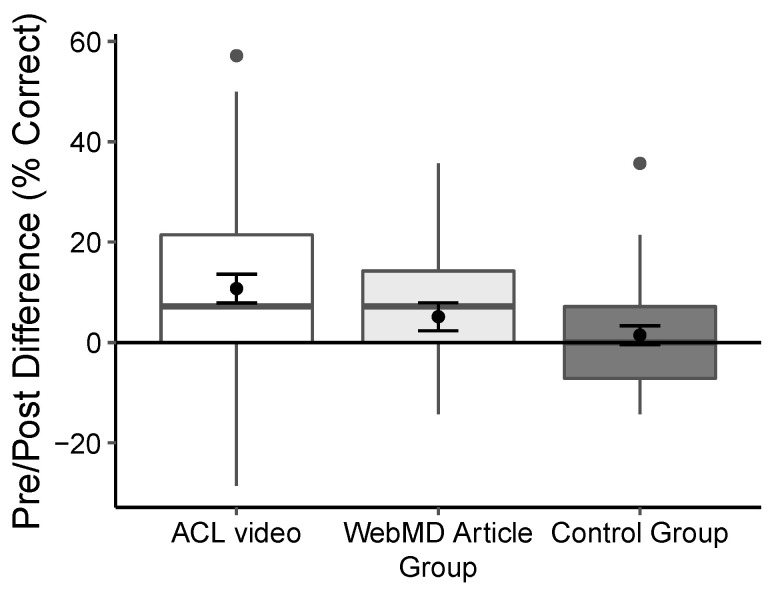
Change in percent correct on overall comprehension between time 1 and time 2 by condition. Black points inside the error bars within the boxplot represent the means and 95% confidence intervals while the black points outside the box represent outliers.

**Table 1 ijerph-18-09092-t001:** Demographic characteristics.

	Animated Video	Web-Based Article	Placebo Control	*p* Value for Group Difference
N	99	71	86	
Gender (% female)	30.3	25.4	29.1	0.77
Age (%)				0.26
18–24	11	8	20	
25–34	41	34	27	
35–44	26	35	37	
45–54	14	14	10	
55 and older	7	8	6	
Years of experience (mean (SD))	6.35 (4.6)	7.93 (5.9)	6.85 (4.6)	0.13
Number of athletes * (mean (SD))	33.6 (21.6)	34.0 (20.8)	38.0 (25.8)	0.38
Gender of athletes (%)				0.23
Boys	48	58	52	
Girls	33	21	26	
Both	18	21	22	
Sports (%)				0.95
Soccer	22	15	19	
Basketball	17	16	15	
Football	14	13	12	
Volleyball	8	3	5	
Lacrosse	1	1	3	
Gymnastics	7	1	5	
Multiple	30	22	27	
Level of sports (%)				0.68
Youth	34	35	38	
Middle School	12	6	9	
High School	13	17	18	
College	3	7	2	
Multiple	37	35	31	
Awareness of IPP ** (% yes)	13	15	16	0.82

* Number of athletes coaches are currently coaching ** IPP = injury prevention programs.

**Table 2 ijerph-18-09092-t002:** Mixed ANOVA results across knowledge factors.

Condition	Degrees of Freedom	F Statistic	*p* Value	Eta-Squared (Partial)	95% CI
Overall Comprehension					
Pre and Post Test	1	57.92	<0.001	0.19	0.12–0.26
Pre and Post Test × Condition	2	14.14	<0.001	0.10	0.05–0.16
Basic ACL Knowledge					
Pre and Post Test	1	31.57	<0.001	0.11	0.06–0.18
Pre and Post Test × Condition	2	2.74	0.07	0.02	0–0.05
Risk Knowledge					
Pre and Post Test	1	31.19	<0.001	0.11	0.06–0.17
Pre and Post Test × Condition	2	9.09	<0.001	0.07	0.02–0.12
Prevention Knowledge					
Pre and Post Test	1	6.92	<0.01	0.03	0–0.07
Pre and Post Test × condition	2	2.34	0.1		0–0.05
Severity Knowledge					
Pre and Post Test	1	31.49	<0.001	0.11	0.06–0.17
Pre and Post Test × condition	2	19.48	<0.001	0.13	0.07–0.2

**Table 3 ijerph-18-09092-t003:** Means and standard deviations of percentage correct for each knowledge factor.

	Pre		Post			
	M	SD	M	SD	Cohen’s d	95% CI
Overall Comprehension						
Animated video	32.77	11.47	43.65	13.65	0.86	0.57–1.16
Web-based article	31.99	13.16	37.12	12.93	0.39	0.06–0.73
Control	34.52	13.93	35.52	14.15	0.09	−0.21–0.40
Basic ACL Knowledge						
Animated video	66.84	35.13	79.29	31.95	0.37	0.09–0.65
Web-based article	66.2	33.36	79.58	33.36	0.38	0.04–0.71
Control	72.35	35.01	76.47	35.01	0.11	−0.19–0.41
Risk Knowledge						
Animated video	36.36	27.39	53.87	21.67	0.71	0.42–1.00
Web-based article	36.15	26.27	46	28.9	0.36	0.02–0.69
Control	37.98	27.13	38.76	27.47	0.03	−0.27–0.33
Prevention Knowledge						
Animated video	30.44	14.36	36.7	18.59	0.38	0.09–0.66
Web-based article	27.93	14.85	30.05	15.33	0.14	−0.19–0.47
Control	31.75	16.28	32.33	17.53	0.03	−0.27–0.34
Severity Knowledge						
Animated video	10.77	15.67	23.57	15.25	0.83	0.54–1.12
Web-based article	13.15	16.41	14.09	16.58	0.06	−0.27–0.39
Control	12.79	16.3	14.34	16.6	0.09	−0.21–0.40

**Table 4 ijerph-18-09092-t004:** Comparisons between conditions at time 2.

Condition	Variable Name	*p* Value *	Cohen’s d	95% CI
Animated video vs. Control				
	Overall Comprehension	<0.01	0.56	0.26–0.86
	Basic ACL Knowledge	1.0	0.08	−0.21–0.37
	Risk Knowledge	<0.01	0.62	0.31–0.91
	Prevention Knowledge	0.29	0.24	−0.05–0.54
	Severity Knowledge	<0.01	0.58	0.28–0.88
Web-based article vs. Control				
	Overall Comprehension	1.0	0.09	−0.23–0.41
	Basic ACL Knowledge	1.0	0.09	−0.23–0.41
	Risk Knowledge	0.21	0.26	−0.06–0.58
	Prevention Knowledge	0.72	−0.14	−0.46–0.18
	Severity Knowledge	1.0	−0.02	−0.33–0.3
Animated video vs. Web-based article				
	Overall Comprehension	<0.01	0.49	0.18–0.8
	Basic ACL Knowledge	1.0	−0.01	−0.32–0.3
	Risk Knowledge	0.15	0.32	0.01–0.62
	Prevention Knowledge	0.06	0.38	0.07–0.69
	Severity Knowledge	<0.01	0.6	0.29–0.91

* *p* value corrected using the Benjamini–Yekutieli adjustment.

**Table 5 ijerph-18-09092-t005:** Means, standard deviations, *t*-tests and Cohen’s *d* values across conditions on implementation feasibility following the interventions. The range of possible scores was 1—completely disagree (low feasibility) to 5—completely agree (high feasibility) across four statements that measured implementation feasibility.

Experiment Group	*N*	Mean	SD	Comparison Group	*t*-Test *p* Value *	Cohen’s d	95% CI
Animated Video	99	4.31	0.71	Control	0.03	0.38	0.09–0.68
Control	86	4.04	0.72	Web-based article	0.03	0.41	0.08–0.72
Web-based article	71	4.30	0.60	Animated Video	1.0	0.01	−0.30–0.32

* *p* value corrected using the Benjamini–Yekutieli adjustment.

## Data Availability

The data presented in this study are available on request to the authors.

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
