# Peer review of "The Effect of a Brief, Web-Based Animated Video for Improving Comprehension and Implementation Feasibility for Reducing Anterior Cruciate Ligament Injury: A Three-Arm Randomized Controlled Trial"

_ijerph, 2021, doi:10.3390/ijerph18179092_

Round 1

Reviewer 1 Report

Thanks to the authors for addressing the final round of comments.

Author Response

Thank you for reviewing our article once again! 

Reviewer 2 Report

Dear Authors,

The manuscript has been improved since I reviewed the 1st version. Congrats for your efforts and commitment. However, there are some important issues not yet addressed.

I previously raised the need of a post hoc correction due to the proposed statistical analysis. It remains the same. So, I argue again about multiple testing. This could be a bias to your results, so please consider the Holm-Bonferroni correction as an option.

You must be aware about the (very strong) statements. No follow up to ensure memory recall or even the actual implement of any action concerning ACL prevention procedures were performed. Thus, the conclusions should be moderated as they still remain theoretical.

Please, see my specific comments in the attached file.

Regards.

Author Response

This is a strong statement, not supported by your current results. Please, moderate the enthusiasm.

  • We have modified the sentence to read: “Overall, these initial results suggest, a brief, web-based, animated video has the potential to be a superior method for informing stakeholders to reduce traumatic injuries in sport.”

How did you qualitatively classified the correlation results? Please, provide reference as I often see this: The correlation coefficients were qualitatively classified as high (>= 0.70), moderate (0.50-0.70), low (0.30-0.50) and weak (<0.30)

Hopkins WG, Marshall SW, Batterham AM, et al. Progressive statistics for studies in sports medicine and exercise science. Medicine and Science in Sports and Exercise 2009; 41: 3–12.

  • We have referenced [48] Fliess at the end of this sentence where the qualitative reliability  heuristics of good and excellent come from.
    • Fleiss, J.L. Design and Analysis of Clinical Experiments; John Wiley & Sons:New York, NY 2011; Volume 73.

Sure, however you must consider the memory bias, as they recall the information immediately after watching the video. The authors should consider this as a limmitation to be discussed.

  • We have included this limitation in the discussion section. “Longer follow-up or delay periods may also improve the design and results to mitigate reduce recall bias.”

The statistical analysis suggests a multiple testing procedure. Please, clarify the measures to avoid biases.

I could not notice any post hoc comparison, except the effect sizes. Please, explain your choice as a correction would help avoiding multiple testing.

  • Multiple testing procedures (i.e., adjustments) have limitations and are not desirable for preliminary/exploratory studies (see Althouse, 2016 reference below). We report what we did, effect sizes, confidence intervals and unadjusted p-values. The readers can use their own judgment about the relative weight of the conclusions. Readers should not be using p values (adjusted or not) to base their decisions off of any way (see Wasserstein and Lazar, 2016).
    • Althouse AD. Adjust for Multiple Comparisons? It's Not That Simple. Ann Thorac Surg. 2016 May;101(5):1644-5. doi: 10.1016/j.athoracsur.2015.11.024. PMID: 27106412.
    • Wasserstein, R. L., & Lazar, N. A. (2016). The ASA's statement on p-values: context, process, and purpose. Amer Stat 70(2):129-133.

By who? A trained rater? Please, clarify.

  • Yes – this has been clarified now.

I do feel that you could make a stronger statemente about your results calculating the MDC. This information would ensure that your results are meaninful answering the stated hypothesis.

  •  We report the effect size and associated confidence intervals.

Now, this is multiple testing. Please, apply the corrections when needed to avoid biases.

  •  See comment above and Althouse, 2016 Ann Thorac Surg for discussion about multiple testing procedures. If readers wanted to calculate the adjusted p value (using the numerous methods) – they have all the information they need (n, means, SD). The Bonferroni corrected alpha is .01250 (.05/4 comparison) and all p’s are less than this. However, this is cautioned and the effect size CI’s should give you a better understanding of the meaningfulness and precision of the effect.

 The assessment was performed once, and just after watching the video. Again, you should consider memory recall bias in this case.

  •  We have modified this sentence to specify “acutely.”

Round 2

Reviewer 2 Report

Dear authors, 

Thank you for your review. However, I feel that the questions raised were not answered. The reader/clinician is not supposed to perform any statistical procedure. Instead, the researcher is responsible for any possible biases due to such issues. I stand for my previous comments though.

Regards.

Author Response

We have now included our methods and results for adjusting the p values for biases associated with multiple comparisons. I believe all other previous comments were addressed. 

Thank you. 

This manuscript is a resubmission of an earlier submission. The following is a list of the peer review reports and author responses from that submission.

Round 1

Reviewer 1 Report

This study investigates an important research area and attempts to measure different approaches to engage coaches to implement injury prevention programs using video education. The authors are congratulated on contributing to this area of need. However, there are some methodological flaws which preclude the recommendation to publish. Overall this study shows that a web-based educational video improves knowledge and understanding of ACL injury and prevention strategies and is superior to control interventions that do not provide the same information. The outcome measures ask questions directly related to content in the video but not the text-based information so it is not a surprise that the video intervention was seen to be superior. This study could have been vastly improved if it was compared to the same text-based information and if the outcomes collected included intention to deliver an IPP. It is not clear if intention to deliver IPP was a question that was asked. The results also show that there was no difference in implementation feasibility between the video and active control group. In such case, the efficacy of using this video to increase uptake in practice is questioned.

Abstract

The outcomes measured are not clearly defined – be consistent with terms e.g. knowledge/understanding in presentation of results.

Introduction

The authors provide a good review of current literature and build a strong case for the type of intervention they have employed. However, the authors could perhaps give more detail of the research on application of video-based learning for more relevant populations i.e. coaches or athletes other similar sports medicine stakeholders. I appreciated that there is not a lot of research in this area (although there are certainly some examples in concussion research) but many of the references given appear to be too far outside the realm of coach/sport specific education that impacts knowledge or behaviour change. A more focused selection would be appreciated here.

Methods

Not enough information is presented on participant recruitment to interpret how representative the sample was. How many were contacted and by what means? Do “panelists” sign up voluntarily or are they contacted to take part based on their demographic? What were the exclusion criteria?

For procedures, it is not clear if the timing of the outcome measure collection was directly before and after the intervention or if there was a time lapse following the intervention? Please clarify the timing of procedures.

The authors did not provide information on how the video intervention was developed. Were topic experts/ behaviour change experts/ coaches and other stakeholders employed to develop the content and was the video pilot tested before the RCT? How/who produced the video?

Line 118 - Please provide more information on the implementation feasibility outcome measure collected. It is not clear how this was determined.

Line 121-123 – The authors state that it was unlikely that participants would be aware of prevention programs prior to the information presentation. From the results, it appears that 15% of the sample were aware of IPPs. Was there any data collected on the number of coaches actually implementing IPPs since this information may be important to consider.

Results

Table 5 – It is difficult for the reader to understand the means presented in this table without an idea of the types of questions asked. Do higher scores indicate more or less feasibility?

Discussion

Line 221 states “The aim of the current study was to evaluate the effects of information format on ACL injury prevention comprehension and implementation feasibility” – however, this is not the comparison that was made since the information and content delivered for video vs. text was not comparable. The text-based information intervention did not contain the information related to risk, prevention and severity knowledge as measured pre- and post. To determine if a brief video is superior to text-based information, it should include the same information but in different format.

It would seem that individuals who were randomised into the third arm (concussion video) may be confused as to why they were completing questionnaires on ACL injury. Was there explanation given as to the aims of the study? Please include the information that was given to participants in the methods.

Author Response

Abstract

Point 1: The outcomes measured are not clearly defined – be consistent with terms e.g. knowledge/understanding in presentation of results.

 Response 1: We have clarified and kept consistent the terminology in the results section of the abstract as well as throughout the manuscript.

Introduction

Point 2:The authors provide a good review of current literature and build a strong case for the type of intervention they have employed. However, the authors could perhaps give more detail of the research on application of video-based learning for more relevant populations i.e. coaches or athletes other similar sports medicine stakeholders. I appreciated that there is not a lot of research in this area (although there are certainly some examples in concussion research) but many of the references given appear to be too far outside the realm of coach/sport specific education that impacts knowledge or behaviour change. A more focused selection would be appreciated here.

 Response 2: We have added an additional paragraph focusing on more sport-specific examples.

Methods

Point 3: Not enough information is presented on participant recruitment to interpret how representative the sample was. How many were contacted and by what means? Do “panelists” sign up voluntarily or are they contacted to take part based on their demographic? What were the exclusion criteria?

Response 3: Further information has been added to the Participants 2.3 section.

Point 4:For procedures, it is not clear if the timing of the outcome measure collection was directly before and after the intervention or if there was a time lapse following the intervention? Please clarify the timing of procedures.

Response 4: The last sentence in the procedures section has been modified to indicate the questions were given immediately following the interventions (i.e., no time lapse). 

Point 5:The authors did not provide information on how the video intervention was developed. Were topic experts/ behaviour change experts/ coaches and other stakeholders employed to develop the content and was the video pilot tested before the RCT? How/who produced the video?

Response 5: Information about the video development is now included in the intervention section 2.4.

Point 6:Line 118 - Please provide more information on the implementation feasibility outcome measure collected. It is not clear how this was determined.

Response 6: The following information was added in the outcomes measures section 2.5: “The 4-items consisted of the root phrase: “An ACL injury prevention program seems” followed by 1) implementable, 2) possible, 3) doable and 4) easy to use. The items were rated on a 5 point Likert-scale with completely disagree to completely agree. The average of the 4-items were used for the analysis.”

Point 7: Line 121-123 – The authors state that it was unlikely that participants would be aware of prevention programs prior to the information presentation. From the results, it appears that 15% of the sample were aware of IPPs. Was there any data collected on the number of coaches actually implementing IPPs since this information may be important to consider.

 Response 7: We did not collect this information. We agree it would be important to know but given the low percent of those aware one could assume that actual use would be lower and not impact the analysis. In addition, the pre/post randomized design should counter any group/baseline differences in pre-existing knowledge.

Results

Point 8:Table 5 – It is difficult for the reader to understand the means presented in this table without an idea of the types of questions asked. Do higher scores indicate more or less feasibility?

 Response 8: In addition to the previous elaboration in the outcomes measures section we also added a brief description in the table legend indicated what the scores indicate.

Discussion

Point 9:Line 221 states “The aim of the current study was to evaluate the effects of information format on ACL injury prevention comprehension and implementation feasibility” – however, this is not the comparison that was made since the information and content delivered for video vs. text was not comparable. The text-based information intervention did not contain the information related to risk, prevention and severity knowledge as measured pre- and post. To determine if a brief video is superior to text-based information, it should include the same information but in different format.

Response 9: We have modified the sentence/aim to more align with the goals. We also modified our claims about information format in the conclusion. We agree that a more “apples to apples” comparison would have led to valid claims about information format. This was not our goal however, as our aim was to assess the videos effectiveness but thought it would be best to have a standard care group in addition to placebo/control group. We wanted to use existing materials for the active comparison group to increase the ecological representativness and more close to the “standard care” or what people may encounter online. Our video also did not address many of the comprehension questions similar to the text-based format. The goal of the comprehension assessment was to be very broad and deep (e.g., what comprehension should represent) and was not designed based on the video-content. The items in the assessment were also based on previous published articles and not developed after the video was created. We also noted in the discussion that there is still a large opportunity for improvement as even in the video group the average comprehension score was only 44% following the intervention.

Point 10: It would seem that individuals who were randomised into the third arm (concussion video) may be confused as to why they were completing questionnaires on ACL injury. Was there explanation given as to the aims of the study? Please include the information that was given to participants in the methods.

Response 10: An explanation of the purpose of the project was communicated in the initial consent page (which we have added more info about in the Procedure 2.2 section).

Reviewer 2 Report

The manuscript’s theme is very interesting and motivating. However, there are several major flaws conducting a RCT that impair the (very optimistic) conclusions. The authors must correct those flaws and moderate their enthusiasm, as many steps were not taken to balance the analysis.

Comments:

The introduction is very clear about the objectives, and the rationale is well evolved. However, there is a major concern: physiotherapy, the core of ACL rehabilitation, was not cited. There are several studies including PT and its protocols for such injury recover.

Did the participants sign an informed consent? Please, clarify.

The allocation concealment was not clear. Please, restructure your text to clarify.

Please, check the CONSORT steps to perform a RCT, and follow the rules. There is no flow chart and no checklist.

“Participants received a small monetary compensation for completing the survey.” Could this procedure bias your results? Consider as a limitation.

“Items that measure ACL knowledge were gathered from peer reviewed published articles [10,32,33].” In my understanding, the items were gathered from distinct questionnaires. Were the psychometric properties measured? “Face validity” seems too vague to ensure construct validity. Please, clarify.

“Results showed moderate to strong positive correlations (r > .67, p<.01), suggesting these measures have strong test-retest reliability and that the participants were taking these questions seriously”. What scale did the authors use to qualitatively classify the level of correlation? Please, clarify.

The timeline is no clear. Please, clarify when the tests were applied. A follow-up was not described. Could this impair the results concerning the level of information’s retaining?

“Participants who responded with incoherent or implausible answers to the demo-125 graphic questions or attention checks were removed from analyses.” Was this a per protocol analysis? Explain why an intention-to-treat analysis was not chosen.

There is no sample size calculation. Please, explain.

The statistical analysis suggests a multiple testing procedure. Also, no normality or homogeneity test was referenced. What level of significance was considered? Please, clarify the measures to avoid those biases.

How the Cohen d test was qualitatively classified? Please, insert the reference.

Table 1 did not show the between-group differences. Were they homogenous?

Table 3 lacks 95% CI info. Please, revise the control’s info.

I could not notice any post hoc comparison, except the effect sizes. Please, explain.

Why did you use the box plot comparison in figure 1 as you used parametrical tests?

“Follow-up independent t-test revealed a statistically significant difference between the text-based and placebo control group conditions (p = .01).” Would you consider this as multiple testing? Also, this procedure was not explained in your stats section.

Consider decrease the number of tables. Plenty of them could be merged into your text, improving the flow. The paragraphs are not fluid to read.

“Subcomponent analyses of ACL injury prevention comprehension revealed greatest improvement within risk knowledge and severity knowledge.” What is this subcomponent analysis? Be careful as you did not perform a component analysis.

“This finding may be due to a ceiling effect since this component consisted of two questions that the majority of participants answered correctly.” As I said before: if the psychometric properties were not previously tested, you may perform a biased analysis.

“Therefore, increasing comprehension of ACL injury prevention among important stakeholders for implementation of IPT in high risk sports will potentially increase the likelihood of IPT adoption and skilled implementation.” Please, moderate your conclusions as you did not assess any type of implementation. There is a long way between knowledge and practice.

Author Response

Comments:

Point 1: The introduction is very clear about the objectives, and the rationale is well evolved. However, there is a major concern: physiotherapy, the core of ACL rehabilitation, was not cited. There are several studies including PT and its protocols for such injury recover.

Response 1: We are not sure we understand the rationale for including information about ACL rehabilitation when the goal is on prevention not rehabilitation. Uptake/adherence of physiotherapy strategies for ACL rehabilitation is an important topic but not related to the core issues in prevention. However, we may be misunderstanding your requestion/point. Are there citation you would like to see here that would rectify your concern?

Point 2: Did the participants sign an informed consent? Please, clarify.

Response 2: Yes, we have added information in the Procedure 2.2 section to clarify this.

Point 3: The allocation concealment was not clear. Please, restructure your text to clarify.

Response 3: Allocation concealment is now described in the Procedure section 2.2.

Point 4: Please, check the CONSORT steps to perform a RCT, and follow the rules. There is no flow chart and no checklist.

Response 4: We have gone through the CONSORT checklist and feel we have addressed all items. We added some additional information about participant flow. A flow chart is recommended but not necessary (as per CONSORT checklist) and given the number of tables/figures we feel it would be more efficient to include this information in text not another figure. All information from the flow chart is presented in text.   

Point 5: “Participants received a small monetary compensation for completing the survey.” Could this procedure bias your results? Consider as a limitation.

Response 5: We do not feel this would bias the results as everyone was given the compensation (regardless of group allocation). In addition, we feel the compensation was appropriate for the amount of time spent on the survey. Most RCT’s compensate participants for their time. If we did not compensate participants for their time we feel this would bias the results more as we’d likely only get VERY interested respondents decreasing the ecological representativeness of typical coaches who have various interest levels.

Point 6: “Items that measure ACL knowledge were gathered from peer reviewed published articles [10,32,33].” In my understanding, the items were gathered from distinct questionnaires. Were the psychometric properties measured? “Face validity” seems too vague to ensure construct validity. Please, clarify.

Response 6: We agree it is important to characterize the psychometric properties. We have described the test-retest characteristics of the used test (which we would argue is most important for knowledge-based assessments). Other psychometric characteristics such as inter-item correlation or grouping (e.g., factor structure) is more difficult to characterize with knowledge-based assessments (see: Taber, K. S. (2018). The use of Cronbach’s alpha when developing and reporting research instruments in science education. Research in Science Education48(6), 1273-1296.). Most of the items used in the ACL understanding battery were from the knowledge assessment used here: Petushek, E. J., Cokely, E. T., Ward, P., Durocher, J. J., Wallace, S. J., & Myer, G. D. (2015). Injury risk estimation expertise: assessing the ACL injury risk estimation quiz. The American Journal of Sports Medicine43(7), 1640-1647; which showed convergent validity evidence. We included a statement about this. We also modified the statement and removed face validity to ensure readers do not get confused about validity evidence.

Point 7: “Results showed moderate to strong positive correlations (r > .67, p<.01), suggesting these measures have strong test-retest reliability and that the participants were taking these questions seriously”. What scale did the authors use to qualitatively classify the level of correlation? Please, clarify.

Response 7: We used Fleiss (1986) and have modified/included the citation here.

Point 8: The timeline is no clear. Please, clarify when the tests were applied. A follow-up was not described. Could this impair the results concerning the level of information’s retaining?

Response 8: We have described in more detail the timeline in the Procedure 2.2 section. We agree that longer-term follow-up would be a great addition but was not part of the aim for this project. We have added a note to this in the limitations section.

Point 9: “Participants who responded with incoherent or implausible answers to the demo-125 graphic questions or attention checks were removed from analyses.” Was this a per protocol analysis? Explain why an intention-to-treat analysis was not chosen.

Response 9: Great question. It was assumed that this would be an intention-to-treat analysis and participants were only excluded if responses indicated they were not attending to the initial screening survey questions or were not a coach of athletes in landing and cutting sports such as soccer and basketball. Failure of major entry criteria can be a reason for exclusion in ITT analyses (but not commonly used). For online studies where panelists may “game the system” for compensation – exclusion of participants based on implausible/incoherent demographic question entry (e.g., checked that they coached soccer in one question but swimming in another) and attention checks (e.g., enter a text phrase on a box) does not seem to be a violation of intention-to-treat analyses. We did not exclude any participants based on “treatment” exposure as in per-protocol approaches. Further information was added about participant flow in the Results section.

Point 10: There is no sample size calculation. Please, explain.

Response 10: Sample size explanation was added to the Participants 2.3 section.

Point 11: The statistical analysis suggests a multiple testing procedure. Also, no normality or homogeneity test was referenced. What level of significance was considered? Please, clarify the measures to avoid those biases.

Response 11: This information has now been provided in the Statistical Analyses 2.6 section. We did test for homogeneity but t and F tests are generally robust against non-normal data too so this shouldn’t be an issue. Many of the homogeneity tests are overly sensitive to non-normal data and are not typically recommended. Given no group differences in baseline measures, non-normality violations should also not bias results. This is also why we chose to use a box-plot (in addition to overlaying mean and 95% CI) so you can see the distribution.

Point 12: How the Cohen d test was qualitatively classified? Please, insert the reference.

Response 12: Reference inserted.

Point 13: Table 1 did not show the between-group differences. Were they homogenous?

Response 13: We did not test between group differences in demographics. Since this is a randomized pre-post design (for learning) demographic differences shouldn’t matter (and unlikely different looking at the data). This is more to characterize the sample for replication.

Point 14: Table 3 lacks 95% CI info. Please, revise the control’s info.

Response 14: Revised

Point 15: I could not notice any post hoc comparison, except the effect sizes. Please, explain.

Response 15: Only effect sizes were calculated for post-hoc pairwise comparisons.

Point 16: Why did you use the box plot comparison in figure 1 as you used parametrical tests?

Response 16: Boxplots are a better representation of the data (shows distributional characteristics – see Weissgerber, T. L., Milic, N. M., Winham, S. J., & Garovic, V. D. (2015). Beyond bar and line graphs: time for a new data presentation paradigm. PLoS Biol13(4), e1002128.) In addition, we plotted the means and CI’s to show the differences from a parametric standpoint.

Point 17: “Follow-up independent t-test revealed a statistically significant difference between the text-based and placebo control group conditions (p = .01).” Would you consider this as multiple testing? Also, this procedure was not explained in your stats section.

Response 17: We have included this information in the Statistical Analysis 2.6 section.

Point 18: Consider decrease the number of tables. Plenty of them could be merged into your text, improving the flow. The paragraphs are not fluid to read.

Response 18: We merged tables 5 and 6 based on your recommendation. If you have other recommendations we would greatly appreciate the specific guidance (e.g., which tables could go in text?).

Point 19: “Subcomponent analyses of ACL injury prevention comprehension revealed greatest improvement within risk knowledge and severity knowledge.” What is this subcomponent analysis? Be careful as you did not perform a component analysis.

Response 19: You are correct – we did not perform component analysis. The subcomponent analysis was just breaking up the Overall Comprehension measure into specific knowledge/comprehension subcomponents (see description in Outcome measures section 2.5). Do you have a recommendation for another word to describe this? We couldn’t think of a better one.

Point 20: “This finding may be due to a ceiling effect since this component consisted of two questions that the majority of participants answered correctly.” As I said before: if the psychometric properties were not previously tested, you may perform a biased analysis.

Response 20: As discussed above and in Taber 2018 – psychometric testing of knowledge assessments is tricky and typical psychometric methods may not be suitable. We do not think that just because there is a ceiling effect on a measure this is a poor measure-  they are just getting it correct (very easy) – which may be good. In this sample, this is the case but in other samples this may not be the case. This is why typical recommendation for knowledge assessments include items that span the difficulty spectrum. We do agree that a more comprehensive psychometric assessment could be warranted but given the sufficient test-retest outcomes we feel these measures/analyses are not biased.

Point 21: “Therefore, increasing comprehension of ACL injury prevention among important stakeholders for implementation of IPT in high risk sports will potentially increase the likelihood of IPT adoption and skilled implementation.” Please, moderate your conclusions as you did not assess any type of implementation. There is a long way between knowledge and practice.

Response 21: We measured feasibility of implementation which is an antecedent to actual implementation (see Weiner et al., 2017). We also say “potentially” and include “likelihood” thus feel these claims are moderated to the extent necessary. We also know that on a theoretical level representative understanding is an antecedent for sustained behavior so feel these claims are justifiable. We also state further down in the paragraph: “Future studies would benefit from recording actual uptake of prevention behaviors in addition to acute intentions or feasibility.”

Round 2

Reviewer 1 Report

Thanks to the authors for clarifying a number of points. While I now understand the aims of the paper, I still do not think it will be clear for the reader why each intervention was chosen. Therefore, I have some additional suggestions.

Be more explicit about what the text-based intervention was from the outset (abstract and aims and methods) i.e. a commonly-accessed web resource on ACL injury and prevention. Using text-based intervention alone implies that you are comparing learning from a video to text which I understand you are not. But this is confusing for the reader. The content in the video was evidence-based but it is not clear if the text based information was too so perhaps this can also be addressed.

The limitations need to state that if another text-based intervention was chosen, the between-group results may be very different. For example, the author mention nih.gov as another highly accessed resource. I believe the quality of information from this resource may be different to one accessed from WebMD so more rationale is needed to the choice of text-based resource.

Author Response

Be more explicit about what the text-based intervention was from the outset (abstract and aims and methods) i.e. a commonly-accessed web resource on ACL injury and prevention. Using text-based intervention alone implies that you are comparing learning from a video to text which I understand you are not. But this is confusing for the reader. The content in the video was evidence-based but it is not clear if the text based information was too so perhaps this can also be addressed.

  • We have modified and changed text-based to web-based article throughout. We have also included more information about the article in the Intervention section 2.4 indicating the evidence-based nature of the content.

The limitations need to state that if another text-based intervention was chosen, the between-group results may be very different. For example, the author mention nih.gov as another highly accessed resource. I believe the quality of information from this resource may be different to one accessed from WebMD so more rationale is needed to the choice of text-based resource.

  • We have included a statement in the limitations section highlighting this as well as the lack of usable web-based resources.

Reviewer 2 Report

Dear authors, thank you for the extensive revision you performed. However, there are still issues to correct. Please, see my further comments below.

Response 4: We have gone through the CONSORT checklist and feel we have addressed all items. We added some additional information about participant flow. A flow chart is recommended but not necessary (as per CONSORT checklist) and given the number of tables/figures we feel it would be more efficient to include this information in text not another figure. All information from the flow chart is presented in text. 

I believe the flowchart improves clarity, Please, include.

Response 13: We did not test between group differences in demographics. Since this is a randomized pre-post design (for learning) demographic differences shouldn’t matter (and unlikely different looking at the data). This is more to characterize the sample for replication.

Between-group differences and proportions may affect this type of study. Please, insert the analysis, make a statement or reference your response.

Response 15: Only effect sizes were calculated for post-hoc pairwise comparisons.

This is not acceptable to infer significant differences, as the multiple testing would be avoided by post hoc corrections. Please, include the p-values, considering the established level of significance.

Response 19: You are correct – we did not perform component analysis. The subcomponent analysis was just breaking up the Overall Comprehension measure into specific knowledge/comprehension subcomponents (see description in Outcome measures section 2.5). Do you have a recommendation for another word to describe this? We couldn’t think of a better one.

When you do not present other options, it is better to make an statement to improve clarity, so the reader can easily interpret your manuscript.

Author Response

Thank you for the prompt feedback. See comments below and changes in manuscript to address your comments. 

I believe the flowchart improves clarity, Please, include.

  • We have included a flow chart.

Between-group differences and proportions may affect this type of study. Please, insert the analysis, make a statement or reference your response.

  • We have included group difference analysis results in both table 1 and in results section.

This is not acceptable to infer significant differences, as the multiple testing would be avoided by post hoc corrections. Please, include the p-values, considering the established level of significance.

  • P values are now included in Table 4.

When you do not present other options, it is better to make an statement to improve clarity, so the reader can easily interpret your manuscript.

  • We have modified the statement about subcomponent analysis to make this more clear it was an analysis of the four facets of comprehension (Discussion sentence 3).